# Mapping awareness of breast and cervical cancer risk factors, symptoms and lay beliefs in Uganda and South Africa

J. Moodley[1,2,3]*, D. Constant[2], A. D. Mwaka[4], S. E. Scott[5], F. M. Walter[6]

1 Women's Health Research Unit, School of Public Health and Family Medicine, Faculty of Health Sciences, University of Cape Town, Observatory, Cape Town, South Africa, 2 Cancer Research Initiative, Faculty of Health Sciences, University of Cape Town, Observatory, Cape Town, South Africa, 3 South African Medical Research Council Gynaecology Cancer Research Centre, Faculty of Health Sciences, University of Cape Town, Observatory, Cape Town, South Africa, 4 Department of Medicine, School of Medicine, College of Health Sciences, Makerere University, Kampala, Uganda, 5 Centre for Oral, Clinical and Translational Sciences, Faculty of Dentistry, Oral and Craniofacial Sciences, King's College London, London, United Kingdom, 6 The Primary Care Unit, Department of Public Health & Primary Care, University of Cambridge, Cambridge, United Kingdom

* Jennifer.moodley@uct.ac.za

## Abstract

### Background

Breast and cervical cancer are leading causes of cancer burden in Sub-Saharan Africa (SSA). We measured breast and cervical cancer symptom and risk factor awareness and lay beliefs in Uganda and South Africa (SA).

### Methods

Between August and December 2018 we conducted a cross-sectional survey of women ≥18 years in one urban and one rural site per country. Households were selected using systematic random sampling, then one woman per household randomly selected to participate. Data were collected by interviewers using electronic tablets customised with the locally validated African Women Awareness of Cancer (AWACAN) tool. This has unprompted questions (testing recall) followed by prompted questions (testing recognition) on risk factor, symptom awareness and lay beliefs for breast and cervical cancer. Mann Whitney and Kruskal Wallis tests were used to compare the association between socio-demographic variables and outcomes. Poisson regression with robust variance was conducted to identify independent socio-demographic predictors.

### Results

Of the 1758 women interviewed, 90.8% had heard of breast and 89.4% of cervical cancer. 8.7% recalled at least one breast risk factor and 38.1% recalled at least one cervical cancer risk factor. 78.0% and 57.7% recalled at least one breast/cervical cancer symptom respectively. Recognition of risk factors and symptoms was higher than recall. Many women were unaware that HPV, HIV, and not being screened were cervical cancer risk factors (23.7%,

**Funding:** Research reported in this article was jointly supported by the Cancer Association of South Africa (CANSA), the University of Cape Town and, the SA Medical Research Council with funds received from the SA National Department of Health, GlaxoSmithKline Africa Non-Communicable Disease Open Lab (via a supporting grant Project Number: 023), the UK Medical Research Council (via the Newton Fund). Authors retained control of the final content of the publication. The funders had no role in study design, data collection and analysis, decision to publish, or preparation of the manuscript.

**Competing interests:** This study was funded in part by GlaxoSmithKline (GSK) Africa Non-Communicable Disease Open Lab (via a supporting grant Project Number: 023). There are no patents, products in development or marketed products to declare. This does not alter our adherence to all the PLOS ONE policies on sharing data and materials.

46.8%, 26.5% respectively). In SA, urban compared to rural women had significantly higher symptom and risk factor awareness for both cancers. In Uganda married women/living with a partner had higher awareness of breast cancer risk factors and cervical cancer symptoms compared to women not living with a partner. Women mentioned several lay beliefs (e.g. putting money in their bra as a breast cancer risk factor).

## Conclusion

We identified gaps in breast and cervical cancer symptom and risk factor awareness. Our results provide direction for locally targeted cancer awareness intervention programs and serve as a baseline measure against which to evaluate interventions in SSA.

## Introduction

Breast and cervical cancer are the leading causes of cancer morbidity and mortality among women in Sub-Saharan Africa (SSA) [1]. Cervical cancer remains the leading cause of female cancer deaths in SSA with mortality rates as high as 40 per 100 000 in Uganda [2]. Breast cancer is the most common cancer in SSA and although incidence rates are lower than those of high-income countries, mortality rates in some SSA countries are similar [2]. For example, age-standardised incidence rates for breast cancer in the United Kingdom (UK) and South Africa (SA) are 94 and 49 per 100 000 women respectively, whilst age-standardised mortality rates in the UK and SA are 14 and 16 per 100 000 women respectively [2]. Most SSA countries do not have cervical or breast cancer screening programs and the majority of cancers are diagnosed symptomatically and at an advanced stage [3]. Late-stage at presentation is a major contributor to low breast and cervical cancer survival rates [4, 5].

Early stage presentation is a key goal of comprehensive cancer policies as it enables more opportunities for curative treatment and improved prognosis among those treated [3]. Prompt help-seeking in the presence of symptoms may lead to earlier-stage diagnoses, and may be influenced by factors such as symptom and risk factor awareness [6–10]. Individual and community understanding of cancer risk and symptoms can include lay beliefs, which may also influence risk reduction and help-seeking behaviours [9, 11].

Understanding the nature and predictors of cancer symptom and risk factor awareness is vital to underpin the development of locally relevant interventions. Programs aimed at raising public and lay health worker awareness of breast and cervical cancer have shown promise in promoting help-seeking behaviour and down-staging disease [6, 12]. Studies have suggested differences in levels of cancer knowledge between African countries. Furthermore, there may also be differences between urban and rural settings within a country [8, 13], underscoring the need for geographically targeted intervention programs.

Most studies measuring breast and cervical cancer awareness in Africa have been conducted in hospital settings among women already diagnosed with the disease [8, 9, 14]. Fewer studies report awareness at a community level among undiagnosed women where levels of cancer symptom and risk factor awareness and beliefs may differ. Furthermore, most studies in SSA, whether hospital or community-based have not used locally validated measures [15]. The aim of this study was to measure breast and cervical cancer symptom and risk factor awareness and lay beliefs in urban and rural settings in Uganda and SA using a culturally relevant, locally validated measurement tool.

## Materials and methods

### Study design

We conducted a community-based cross-sectional survey of women aged 18 years and older in one urban and one rural site in Uganda and SA (total of 4 sites). Uganda is a low-income country with a population of 34.9 million [16]. Both the urban and rural study sites were in Northern Uganda, the poorest of the four regions in Uganda. SA is a middle-income country with a population of 58.8 million [17]. Our urban SA study site was in the Western Cape Province, one of the wealthier of the nine provinces in the country, whereas our rural site was in the Eastern Cape Province which has very low levels of wealth [18].

### Public involvement

The African Women Awareness of Cancer (AWACAN) project was set-up to improve timely cancer diagnosis [19]. At the start of the AWACAN project, community-based project advisory committees (PACs) were set up in each county. The PACs included local government councillors, traditional leaders as well as representatives from the cancer registry and public sector health services (health service providers and managers). The AWACAN breast and cervical cancer tool used in the survey was developed with inputs from PAC members. In this study, PAC members assisted in gaining access to communities, recruiting local women as field workers in each site and provided guidance on field worker safety. AWACAN results have been discussed with PAC members and field workers in a series of meetings.

### Participants

At each site, households were selected using systematic random sampling. The name of each woman in the household was written on a separate piece of paper, folded and placed in a "hat". Random selection of a participant was then done by picking one from the set of folded papers. If the woman selected did not want to participate or there were no women in the household, fieldworkers moved to the next marked household, noting the reason for non-participation. If the selected woman from the household was away, fieldworkers returned for an interview later. Women unable to speak either Acholi, isiXhosa or English; women with a history of breast or cervical cancer and; woman younger than 18 years were excluded.

### Data collection

Data were collected by trained interviewers using a hand-held electronic tablet customised with the AWACAN breast and cervical cancer tool. Details of the content, development and validation of the tool, are described elsewhere [15]. Women were interviewed face-to-face in a private space in their homes, either in their local language (isiXhosa or Acholi) or in English depending on their preference.

**Measures of socio-demographics.** We collected the following socio-demographic details: age, relationship status, highest educational level, work status i.e. whether doing paid work, and information on access to assets. Asset variables in the questionnaire were based on previous research in SA using General Household Surveys [20]. We constructed a composite asset index to describe socioeconomic status using principal component analysis (PCA) on a model that included asset variables with a positive correlation at $p \leq 0.01$. Only five variables satisfied this condition (access to piped water, electricity, radio, television and the internet). To ensure consistency between countries, we used the same set of variables for both SA and Uganda, treating each country individually. The asset index was then divided into terciles, as it did not sufficiently discriminate between households to allow for more than three levels. To check the

validity of the asset index terciles, we tested for association with other key variables, namely paid work and highest education level; all associations were significant at p<0.001.

**Measures of cancer awareness.** We measured cancer awareness by asking whether the participant had heard of breast/cervical cancer and whether they knew of any family members, friends or neighbours who had breast/cervical cancer. Risk factor and symptom awareness was assessed using an open/unprompted question followed by closed/prompted question format as laid out in the AWACAN breast and cervical cancer tool [15]. As risk factor questions related specifically to either breast or cervical cancer, these questions were posed only to participants that had heard of breast or cervical cancer. We asked all participants about symptom awareness as these questions as these were framed around whether a symptom was viewed as a sign of something serious. Breast cancer risk factor open questions (measuring recall) read "Please can you name as many things as you can think of that could increase any woman's chances of getting breast cancer". Open questions for symptoms read "Name as many symptoms or signs of breast cancer as you can think of". Closed questions (measuring recognition) asked "Could any of the following increase any woman's chances of getting breast cancer" (13 items) and "Do you think the following could be signs of something serious or that something is wrong, such as breast cancer" (15 items) [15]. We used the AWACAN tool open and closed question format to measure recall and recognition of cervical cancer risk factors and symptoms. There were 11 closed questions assessing cervical cancer risk factor recognition and 11 questions assessing symptom recognition [15]. The AWACAN questionnaire includes prompted lay belief items: 6 breast cancer risk factors, 1 breast cancer symptom, 4 cervical cancer risk factors and 1 cervical cancer symptom [15]. Additional lay beliefs were identified during unprompted questioning.

For open questions, women scored '1' if they mentioned a risk factor or symptom that corresponded to the list of closed questions, and '0' if this was not mentioned. Scoring of open responses was done by two investigators (DC and JM). For closed questions measuring recognition women scored '1' for 'Yes' and '0' for 'No' or 'Don't know'. Scores for each response to the closed questions were summed to form separate knowledge scores for breast cancer risk factors (0–13), breast cancer symptoms (0–15), cervical cancer risk factors (0–11) and symptoms (0–11). Higher scores indicated better recognition of symptoms and risk factors and greater endorsement of lay beliefs. The median and the interquartile range (IQR) are reported for symptom and risk factor scores. Similar to other studies we dichotomised scores for risk factor and symptom awareness using the median as a cut-off for low and high awareness [21–23]. The median was classified as low awareness to achieve an even distribution between groups.

## Ethics

Ethics approval for the study was obtained from University of Cape Town, Faculty of Health Sciences Human Research Ethics Committee (HREC 544/2016), the Lacor Hospital Institutional Research Ethics Committee (LHIREC 027/11/2016) and the Ugandan National Council of Science and Technology (HS60ES). Consent forms were translated and administered in the participants' preferred language. Written informed consent was obtained from all participants.

## Sample size and analysis

To detect differences in awareness scores of at least 10% between groups for the various comparisons (e.g. between countries, by urban/rural location, by age groups) assuming a two-sided two-sample test with alpha = 0.05, a standard deviation of 0.2 and allowing for a 15% non-response rate, we aimed to recruit 400 women at each site.

We analyzed the data using Stata version 15, using valid percentages to report socio-demographic data. Medians and interquartile ranges (IQR) were calculated for continuous variables. As Ugandan and SA socio-demographic indicators showed significant differences, we did the bivariate and multivariable analysis by country to best demonstrate these context specific effects. As none of the scores were normally distributed, we compared scores according to socio-demographic factors in bivariate analyses using Mann Whitney and Kruskal Wallis tests as appropriate, using a significance level of $p \leq 0.01$. Poisson regression with robust variance was conducted to identify independent socio-demographic predictors for risk factor and symptom awareness. Prevalence ratios (PR) and 95% confidence intervals are reported. The regression models included the following variables: location, age, relationship status, highest education level, paid work and asset index.

## Results

Between August and December 2018, a total of 1941 households were visited in Uganda and SA, and 1758 participants recruited. The overall refusal rate was 6.3% and was highest in the urban Ugandan site (13.7%). S1 Appendix Participant recruitment provides details of site recruitment, reasons for refusal and ineligibility, and final enrolment numbers.

The results are presented in 3 sections: Section 1 outlines the Participant profile. The results on awareness of risk factors and symptoms as well as lay beliefs are presented under Section 2: Breast cancer and Section 3: Cervical cancer

### Section 1: Participant profile

The median age of respondents was 34 years (IQR 26–47). Participant characteristics by country and site are outlined in Table 1. A higher proportion of women in rural Uganda were married and had no or incomplete primary schooling compared to other sites. Overall, 66.3% of participants did no paid work, with the highest rate in rural SA (91.8%).

Asset distribution differed by site. Women in urban SA had the most assets (73.3% at upper tercile), whilst those in rural SA had the fewest assets (56.1% at lower tercile). S2 Appendix provides details on participants' assets by site and by country. Overall, two-thirds of women had access to a radio, half had access to television and just under half had access to the internet, with differences between countries and urban/rural sites. Most women (90.8%%) had heard of breast cancer; of those, 30.3% knew of someone with the disease. Most women (89.4%) had heard of cervical cancer, and of those, almost a third (29.0%) knew of someone with the disease.

### Section 2: Breast cancer

**Breast cancer risk factor awareness.** Risk factor recall was low in all sites with only 8.7% of participants among those that had heard of breast cancer (n = 1596) able to recall at least one risk factor (S3 Appendix). Recognition of risk factors was better than recall with 95.2% of women able to recognize at least 1 of 13 risk factors. However, most women could only recognize less than half of the 13 risk factors (median risk recognition score = 4, IQR 3–7). Recall and recognition of individual risk factor and symptoms by site are provided in S3 Appendix. Overall, the most commonly recognized risk factor was a past history of breast cancer (52.9%.) and the least commonly recognized was doing little physical activity (24.1%). Fig 1A depicts the proportion of women recognizing each risk factor by site.

In SA, women living in an urban location, those aged between 30 and 49, those with paid work, and women with the highest asset index assets had greater awareness of breast cancer risk factors (Table 2, bivariate analysis). On regression urban location remained a significant

**Table 1. Participant profile.**

| | South Africa | | Uganda | | Total |
|---|---|---|---|---|---|
| | Urban | Rural | Urban | Rural | |
| | n (%) n = 445 | n (%) n = 428 | n (%) n = 458 | n (%) n = 427 | n (%) n = 1758 |
| **Age (years)** | | | | | |
| 18–29 | 130 (30.4) | 106 (25.2) | 216 (47.2) | 186 (44.0) | 638 (36.9) |
| 30–49 | 221 (51.6) | 170 (40.4) | 174 (38.0) | 146 (34.5) | 711 (41.1) |
| ≥ 50 | 77 (18.0) | 145 (34.4) | 68 (14.9) | 91 (21.5) | 381 (22.0) |
| Median (IQR) | 36 (28–45) | 40 (29–54) | 30 (24–40) | 32 (23–46) | 34 (26–47) |
| **Relationship status** | | | | | |
| Married/Living with a partner | 185 (41.8) | 200 (46.7) | 300 (65.5) | 314 (73.5) | 999 (56.9) |
| No partner/not living with partner | 224 (50.6) | 174 (40.7) | 36 (7.9) | 17 (4.0) | 451 (25.7) |
| Separated/Divorced/Widowed | 34 (7.7) | 54 (12.6) | 122 (26.6) | 96 (22.5) | 306 (17.4) |
| **Highest educational level** | | | | | |
| No schooling & primary incomplete | 22 (5.0) | 162 (38.0) | 171 (37.3) | 328 (76.8) | 683 (39.0) |
| Primary complete & secondary incomplete | 196 (44.3) | 178 (41.8) | 167 (36.5) | 84 (19.7) | 625 (35.6) |
| Secondary complete or more | 224 (50.7) | 86 (20.2) | 120 (26.2) | 15 (3.5) | 445 (25.4) |
| **Paid work** | | | | | |
| Yes | 226 (51.6) | 35 (8.2) | 217 (47.6) | 110 (25.8) | 588 (33.7) |
| No | 212 (48.4) | 391 (91.8) | 239 (52.4) | 317 (74.2) | 1159 (66.3) |
| **Asset index** | | | | | |
| Upper tercile | 326 (73.3) | 6 (1.4) | 253 (55.2) | 70 (16.4) | 655 (37.3) |
| Middle tercile | 79 (17.8) | 182 (42.5) | 122 (26.6) | 210 (49.2) | 593 (33.7) |
| Lower tercile | 40 (9.0) | 240 (56.1) | 83 (18.1) | 147 (34.4) | 510 (29.0) |
| **Ever heard of breast/cervical cancer** | | | | | |
| Heard of breast cancer | 406 (91.2) | 342 (79.9) | 445 (94.4) | 403 (94.3) | 1596 (90.8) |
| Heard of cervical cancer | 366 (82.3) | 342 (79.9) | 452 (98.7) | 411 (96.3) | 1571 (89.4) |

There were missing data for

Age: South Africa Urban; n = 17, Rural; n = 7. Uganda Rural; n = 4. Total; n = 28.

Relationship status: South Africa Urban; n = 2. Total; n = 2.

Highest educational level: South Africa Urban; n = 3, Rural; n = 2. Total; n = 5.

Paid work: South Africa Urban; n = 7, Rural; n = 2. Uganda Urban = 2. Total; n = 11.

predictor of greater awareness of breast cancer risk factors (PR 1.35, 95% CI 1.07–1.72). Women aged between 30 and 49 years also had significantly higher risk factor awareness compared to women younger than 30 years (PR 1.28; 95% CI 1.03–1.58) (S4 Appendix). In Uganda, relationship status, highest educational level and asset index were associated with risk factor recognition on bivariate analysis (Table 3). On multivariable analyses the only factor that remained significant was relationship status, women not living with a partner/having no partner had significantly lower risk awareness compared to women that were married/living with a partner (PR 0.37; 95% CI 0.19–0.73) (S5 Appendix)

**Breast cancer symptom awareness.** Breast cancer symptom recall and recognition was high (S3 Appendix Recalled and recognized breast cancer risk factors and symptoms by site). Overall, 78.0% of women could recall at least one breast cancer symptom, most commonly a breast lump or thickening. Breast cancer symptom recognition was higher than recall with 99.0% of participants recognizing at least one of 15 symptoms and an overall median symptom recognition score of 14 (IQR 11–15). A breast lump was the most recognized symptom (90.8% of all participants); Fig 1B shows the proportion of women recognizing each symptom by site.

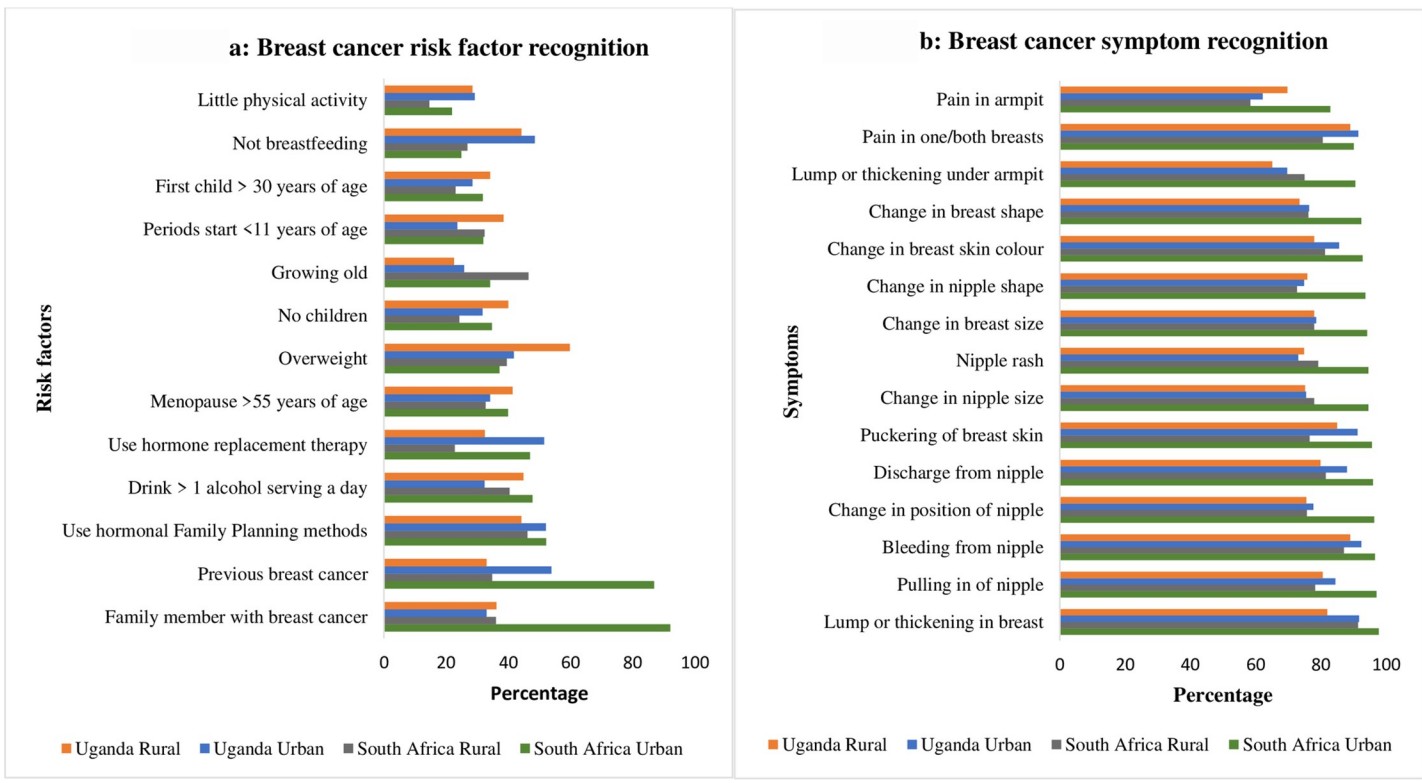

**Fig 1. Breast cancer risk factor and symptom recognition by site.**

In SA, location, highest education level, paid work and asset index) was associated with symptoms awareness (Table 2, bivariate analysis). On regression analysis women living in the urban site were significantly more likely to know about breast cancer symptoms compared to rural women (PR 2.83; 95% CI 2.24–3.57), as were women who had completed secondary education or more compared to those with no or primary incomplete educational levels (PR 1.41; 95% CI 1.04–1.91) (S4 Appendix). Women doing paid work were less aware of breast symptoms compared to those not doing paid work (PR 0.84; 95% CI 0.71–0.98) (S4 Appendix). For Uganda, none of the socio-demographic factors were associated with breast cancer symptom awareness on bivariate analysis (Table 3), but on multivariable analysis older age was significantly associated with greater symptom awareness (women aged between 30 and 49 years compared to women younger that 30 years old PR 1.37; 95% CI 1.15–1.63 and, women 50 years and older compared to women younger than 30 years old PR 1.32; 95% CI 1.04–1.69) (S5 Appendix).

**Breast cancer lay beliefs.** On unprompted questioning, some women (21.1%) mentioned putting money in their bra as a possible breast cancer risk factor. This lay belief was strongly endorsed in prompted questioning, particularly among women in urban SA (97.8%). Other lay beliefs relating to undergarments were also strongly endorsed in prompted questions (S6 Appendix Unprompted and prompted breast and cervical cancer lay beliefs). Lay beliefs related to being bewitched were not commonly held, mentioned by only 0.8% of all participants in unprompted questioning and endorsed by 22.2% in prompted questioning.

### Section 3: Cervical cancer

**Cervical cancer risk factor awareness.** Of the 1571 participants that had heard of cervical cancer, 38.1% were able to recall at least one risk factor. In Uganda, having multiple sexual

**Table 2. Socio-demographic predictors of breast risk factor and symptom awareness in South Africa.**

| | Recognized breast cancer risk factors N = 748[*] | | Recognized breast cancer symptoms N = 873[**] | |
|---|---|---|---|---|
| | Median (Interquartile Range) | Test statistic, p-value | Median (Interquartile Range) | Test statistic, p-value |
| **Location** | | z = 5.883, p<0.001 | | z = 12.839, p<0.001 |
| Urban | 5 (3–9) | | 15 (14–15) | |
| Rural | 4 (2–6) | | 13 (10–14) | |
| **Age** | | $chi^2$ = 12.637, p = 0.002 | | $chi^2$ = 3.760, p = 0.153 |
| 18–29 | 4 (2–7) | | 14 (12–15) | |
| 30–49 | 5 (3–8) | | 14 (12–15) | |
| $\geq$ 50 | 4 (3–6) | | 14 (11–15) | |
| **Relationship status** | | $chi^2$ = 3.128, p = 0.209 | | $chi^2$ = 4.500, p = 0.105 |
| Married/Living with a partner | 4 (3–7) | | 14 (12–15) | |
| No partner/not living with partner | 4 (2–7) | | 14 (12–15) | |
| Separated/Divorced/Widowed | 4 (2–6) | | 14 (10–15) | |
| **Highest educational level** | | $chi^2$ = 4.427, p = 0.109 | | $chi^2$ = 42.051, p<0.001 |
| No schooling to primary incomplete | 4 (2–7) | | 13 (10–14) | |
| Primary complete to secondary incomplete | 4 (2–7) | | 14 (12–15) | |
| Secondary complete or more | 4 (3–8) | | 15 (13–15) | |
| **Paid work** | | z = -5.302, p<0.001 | | z = -5.691, p<0.001 |
| Yes | 5 (3–9) | | 15 (13–15) | |
| No | 4 (2–6) | | 14 (11–15) | |
| **Asset index** | | $chi^2$ = 12.158, p = 0.002 | | $chi^2$ = 66.029, p<0.001 |
| Upper Tercile | 5 (3–8) | | 15 (13–15) | |
| Middle Tercile | 4 (2–7) | | 14 (11–15) | |
| Lower Tercile | 4 (2–7) | | 13 (10.5–15) | |

[*]N = [number enrolled–number that had not heard of breast cancer]

[**] N = number enrolled.

Where 2 categories of independent variables are compared: z statistic is reported for Mann-Whitney two sample test.

Where 3 categories of independent variables are compared: chi-squared statistic is report for Kruskal-Wallis rank test.

partners was the most commonly recalled risk factor. In urban SA having unprotected sex was the most commonly recalled risk factor, whilst in rural SA having other sexually transmitted diseases was most commonly recalled. Not going for regular screening and having an HPV infection were very poorly recalled, with <1% of all participants mentioning either of these risk factors. S7 Appendix gives information on recall and recognition of each cervical cancer risk factor and symptom by site.

Recognition of risk factors was better than recall, with 99.1% recognizing at least 1 of 11 risk factors. Overall, the median score for risk factor recognition was 8 (IQR 6–9), and the most commonly risk factor recognized was having many sexual partners. Of the participants 21.8% in Uganda and 26.1% in SA, did not recognize that getting an HPV infection was a risk factor for cervical cancer (Fig 2). Some women did not recognize that not going for regular screening was a risk factor for cervical cancer, with considerable variation across sites (7.9% in urban SA, 26.6% in rural SA, 37.8% in urban Uganda and 30.7% in rural Uganda).

For SA, geographical location and paid work were significantly associated with cervical cancer risk awareness on bivariate analysis (Table 4). Only living in an urban location remained significantly associated with higher cervical cancer risk factor awareness in regression analysis (PR 1.55; 95% CI 1.22–1.99) (S8 Appendix). For Uganda, location, highest educational level and asset index were associated with cervical cancer risk recognition on bivariate analysis

**Table 3. Socio-demographic predictors of breast risk factor and symptom awareness in Uganda.**

| | Recognized breast cancer risk factors N = 848* | | Recognized breast cancer symptoms N = 885** | |
|---|---|---|---|---|
| | Median (Interquartile Range) | Test statistic, p-value | Median (Interquartile Range) | Test statistic, p-value |
| **Location** | | z = 0.882, p = 0.378 | | z = -1.171, p = 0.242 |
| Urban | 5 (2–7) | | 13 (11–15) | |
| Rural | 5 (3–7) | | 13 (11–15) | |
| **Age** | | chi² = 6,865, p = 0.032 | | chi² =, 7.813 p = 0.020 |
| 18–29 | 4 (2–7) | | 13 (10–14) | |
| 30–49 | 5 (3–8) | | 13 (11–15) | |
| ≥ 50 | 5 (3–7) | | 13 (9–15) | |
| **Relationship status** | | chi² = 17.698, p<0.001 | | chi² = 8.291, p = 0.016 |
| Married/Living with a partner | 5 (3–7) | | 13 (11–15) | |
| No partner/not living with partner | 2 (1–5) | | 12 (10–14) | |
| Separated/Divorced/Widowed | 5 (3–7) | | 13 (9–14) | |
| **Highest educational level** | | chi² = 13.686, p = 0.001 | | chi² =, 6.817 p = 0.033 |
| No schooling to primary incomplete | 5 (3–7) | | 13 (10–15) | |
| Primary complete to secondary incomplete | 4 (2–7) | | 13 (10–14) | |
| Secondary complete or more | 4 (2–6) | | 13 (12–15) | |
| **Paid work** | | z = 2.114, p = 0.035 | | z = 0.057, p = 0.954 |
| Yes | 4 (2–7) | | 13 (10–15) | |
| No | 5 (3–7) | | 13 (10–15) | |
| **Asset index** | | chi² = 9.257, p = 0.010 | | chi² = 4.203, p = 0.122 |
| Upper Tercile | 4 (2–7) | | 13 (11–15) | |
| Middle Tercile | 5 (3–7) | | 13 (10–15) | |
| Lower Tercile | 5 (3–7) | | 13 (10–14) | |

*N = [number enrolled–number that had not heard of breast cancer]

** N = number enrolled.

Where 2 categories of independent variables are compared: z statistic is reported for Mann-Whitney two sample test.

Where 3 categories of independent variables are compared: chi-squared statistic is report for Kruskal-Wallis rank test.

(Table 5). Factors that remained significant on regression analysis were secondary or higher educational level compared to no schooling/incomplete primary education (PR 0.66; 95% 0.46–0.96); paid work (PR 0.82; 95% CI 0.67–0.99) (S9 Appendix).

**Cervical cancer symptom awareness.** In total 57.7%, of women were able to recall at least one symptom of cervical cancer. Recognition of symptoms was higher than recall, with 97.9% of participants able to recognize at least one symptom. Overall, the most commonly recognized symptom was a smelly vaginal discharge (89.9%) and more than 80.6% of women recognized unusual vaginal bleeding as a symptom. The least commonly recognized symptom overall was persistent diarrhea (35.0%) (Fig 2B).

In SA, women in the urban site and those doing paid work had significantly higher cervical cancer symptom awareness on bivariate analysis (Table 4). After adjusting for all included variables, factors that were significantly associated with higher cervical cancer risk recognition in SA were living at an urban site (PR 1.79; 95% CI 1.41–2.27), older age (women aged 30–50 years, PR 1.37; 95% CI 1.10–1.71 and women ≥50 years and older PR 1.47; 95% CI 1.13–1.92, each compared to women younger than 30 years) and, women with the least compared to the most assets (PR 1.34; 95% CI 1.05–1.70) (S8 Appendix). In Uganda, women with no partner/ not living with a partner had significantly lower symptom awareness compared to women that were married or living with a partner (PR 0.60; 95% CI 0.37–0.98) (S9 Appendix).

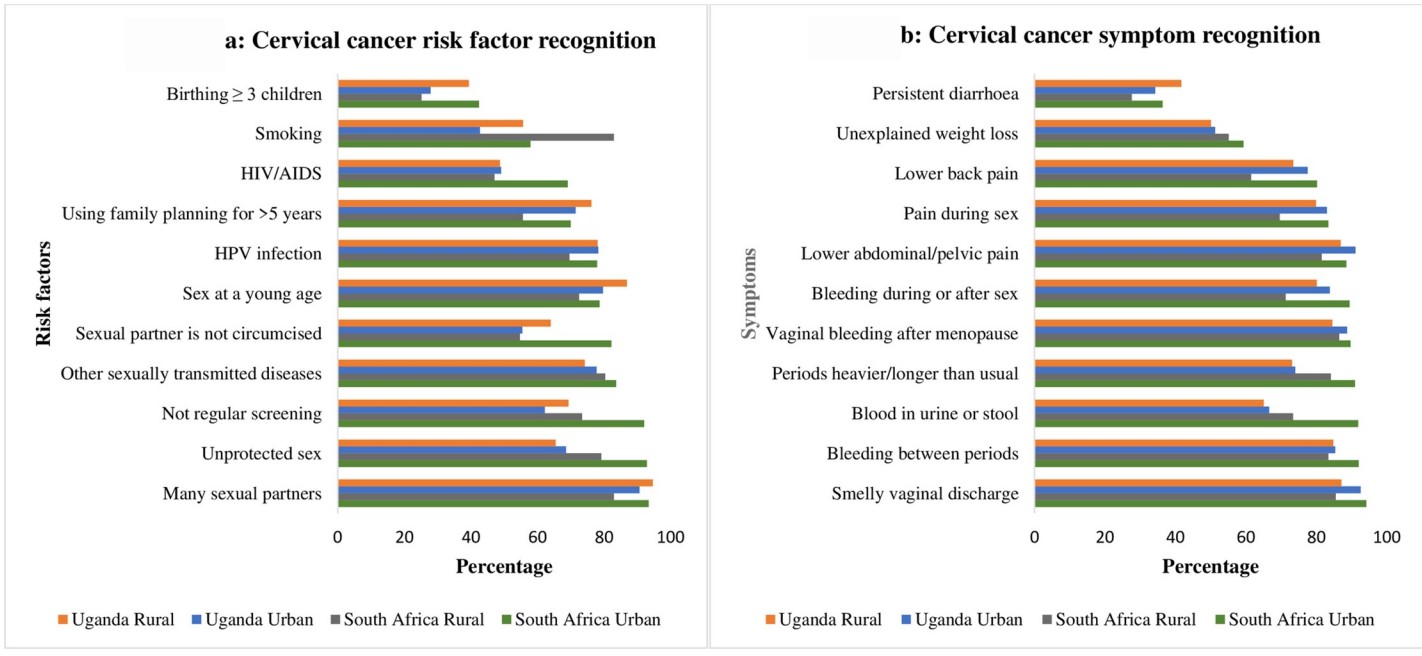

**Fig 2. Cervical cancer risk factor and symptom recognition by site.**

**Cervical cancer lay beliefs.** Overall, poor personal hygiene was most frequently raised as a risk factor for cervical cancer both on prompted (92.9%) and unprompted questioning (33.7%), although fewer women in rural SA raised or endorsed this belief (S6 Appendix). When prompted, most women also endorsed vaginal insertion of herbs/creams/object as a possible risk factor (85.4% of all women). As with breast cancer, lay beliefs related to being bewitched were not commonly held, mentioned by <0.5% of all participants in unprompted questioning and endorsed by 22.3% in prompted questioning.

## Discussion

This is the first study measuring community level breast and cervical cancer awareness across diverse settings in SSA using a locally validated questionnaire. We found that, although most women had heard of both cancers in urban and rural settings in SA and Uganda, knowledge of risk factors and symptoms was low, particularly when measured in an unprompted mode (testing recall). In line with findings from previous studies [10, 14, 24, 25], we found that risk factor and symptom awareness was higher with prompted questioning. With prompted questioning, participants were provided with cues (the questions) and limited response options, thus making the recognition task easier than the recall task of unprompted questioning. In addition the prompted mode of questioning is prone to guessing, which could in part account for higher scores.

Awareness of risk factors for breast cancer was very low in all four of our study sites, both on unprompted and prompted questioning. Similar to other studies in low-and middle-income African settings [8, 9, 14, 26], we found poor levels of knowledge for factors which are responsive to interventions such as physical exercise and being overweight. Information on these risk factors need to be included in cancer awareness programs. In addition, as these risk factors are common to other non-communicable diseases (NCDs) such as cardiovascular disease and diabetes, NCD behavioral intervention programs and service platforms could be used

**Table 4. Socio-demographic predictors of cervical cancer risk factor and symptom awareness in South Africa.**

| | Recognized cervical cancer risk factors N = 708[*] | | Recognized cervical cancer symptoms N = 873[**] | |
| --- | --- | --- | --- | --- |
| | Median (Interquartile Range) | Test statistic, p-value | Median (Interquartile Range) | Test statistic, p-value |
| **Location** | | z = 5.717, p<0.001 | | z = 6.916, p = <0.001 |
| Urban | 9 (7–10) | | 9 (8–11) | |
| Rural | 8 (6–9) | | 8 (6–10) | |
| **Age** | | chi$^2$ = 1.675 p = 0.433 | | chi$^2$ = 6.050, p = 0.049 |
| 18–29 | 8 (6–10) | | 9 (7–10) | |
| 30–49 | 8 (7–10) | | 9 (8–10) | |
| ≥ 50 | 8 (7–9) | | 9 (7–10) | |
| **Relationship status** | | chi$^2$ = 3.033, p = 0.220 | | chi$^2$ = 2.194, p = 0.334 |
| Married/Living with a partner | 8 (7–10) | | 9 (7–10) | |
| No partner/not living with partner | 8 (6–10) | | 9 (7–10) | |
| Separated/Divorced/Widowed | 7.5 (6–9) | | 9 (6.5–10) | |
| **Highest educational level** | | chi$^2$ = 3.289, p = 0.193 | | chi$^2$ = 2.753, p = 0.252 |
| No schooling to primary incomplete | 8 (6–10) | | 9 (7–10.5) | |
| Primary complete to secondary incomplete | 8 (7–10) | | 9 (7–10) | |
| Secondary complete or more | 8 (6–10) | | 9 (7–10) | |
| **Paid work** | | z = -3.177, p = 0.002 | | z = -3.299 p = 0.001 |
| Yes | 9 (7–10) | | 9 (8–11) | |
| No | 8 (6–9) | | 9 (7–10) | |
| **Asset index** | | chi$^2$ = 8.422, p = 0.015 | | chi$^2$ = 8.954, p = 0.011 |
| Upper Tercile | 9 (7–10) | | 9 (8–10) | |
| Middle Tercile | 8 (6–10) | | 9 (7–10) | |
| Lower Tercile | 8 (6–10) | | 9 (7–10) | |

[*]N = [number enrolled–number that had not heard of cervical cancer]

[**] N = number enrolled.

Where 2 categories of independent variables are compared: z statistic is reported for Mann-Whitney two sample test.

Where 3 categories of independent variables are compared: chi-squared statistic is report for Kruskal-Wallis rank test.

to educate women about the link between these factors and breast cancer [3]. Adapting existing interventions to include additional messaging could be an efficient and cost-effective approach to addressing low breast cancer risk factor awareness.

Most women were aware that a lump was a possible symptom of breast cancer, whereas armpit changes were not well recognized. Again, these findings have been reported from studies in similar settings [8, 14], and have also been associated with delayed presentation [7, 27, 28]. Differences in levels of awareness of different symptoms could be related to the emphasis placed by cancer education initiatives on a breast lump compared to other symptoms. As one in six women with breast cancer may present with a non-lump symptom [27], breast cancer awareness programs need to also highlight these less well-known symptoms in order to contribute to more timely diagnosis of breast cancer.

Cervical cancer has been recognized as a major public health problem in SA and Uganda, with both countries offering public sector prevention programs that include HPV vaccination (SA and Uganda) and cervical cancer screening (SA) [29, 30]. Of concern is the finding that in our study many women in rural SA and urban and rural Uganda did not identify lack of screening as a risk factor for cervical cancer. Poor screening awareness has also been reported elsewhere in Africa [31, 32]

**Table 5. Socio-demographic predictors of cervical cancer risk factor and symptom awareness in Uganda.**

| | Recognized cervical cancer risk factors N = 863[*] | | Recognized cervical cancer symptoms N = 885[**] | |
|---|---|---|---|---|
| | Median (Interquartile Range) | Test statistic, p-value | Median (Interquartile Range) | Test statistic, p-value |
| **Location** | | z = 3.179, p = 0.002 | | z = -0.129, p = 0.898 |
| Urban | 7 (5–9) | | 9 (7–10) | |
| Rural | 8 (6–9) | | 9 (6–10) | |
| **Age** | | chi$^2$ = 1.875, p = 0.392 | | chi$^2$ = 5.847 p = 0.054 |
| 18–29 | 8 (6–9) | | 9 (7–10) | |
| 30–49 | 8 (6–9) | | 9 (7–10) | |
| $\geq$ 50 | 7 (6–9) | | 9 (6–10) | |
| **Relationship status** | | chi$^2$ = 6.512, p = 0.039 | | chi$^2$ = 8.956, p = 0.011 |
| Married/Living with a partner | 8 (6–9) | | 9 (7–10) | |
| No partner/not living with partner | 6.5 (4–9) | | 8 (6–9) | |
| Separated/Divorced/Widowed | 8 (5–9) | | 9 (6–10) | |
| **Highest educational level** | | chi$^2$ = 18.840, p<0.001 | | chi$^2$ = 1.430, p = 0.489 |
| No schooling to primary incomplete | 8 (6–10) | | 9 (6–10) | |
| Primary complete to secondary incomplete | 8 (6–9) | | 9 (7–10) | |
| Secondary complete or more | 7 (5–8) | | 9 (7–10) | |
| **Paid work** | | z = 2.351, p = 0.019 | | Z = 1.805, p = 0.071 |
| Yes | 7 (6–9) | | 9 (6–10) | |
| No | 8 (6–9) | | 9 (7–10) | |
| **Asset index** | | chi$^2$ = 16.046, p<0.001 | | chi$^2$ = 3.976, p = 0.137 |
| Upper Tercile | 7 (5–9) | | 9 (7–10) | |
| Middle Tercile | 8 (6–10) | | 9 (7–10.5) | |
| Lower Tercile | 8 (5–10) | | 9 (6–10) | |

[*]N = [number enrolled–number that had not heard of cervical cancer]

[**] N = number enrolled.

Where 2 categories of independent variables are compared: z statistic is reported for Mann-Whitney two sample test.

Where 3 categories of independent variables are compared: chi-squared statistic is report for Kruskal-Wallis rank test.

The association between HIV and cervical cancer is well established [33, 34] and most screening programs offer more frequent screening for HIV infected women [29, 35] In SA cervical cancer screening guidelines have been incorporated into HIV services for HIV infected women [29]. Despite these efforts, almost half the women in our study did not recognize that HIV was a risk factor for cervical cancer. Efforts to improve community awareness around the benefits of screening and about the link between HIV and cervical cancer may help increase screening coverage and reduce the prevailing high cervical cancer incidence.

Despite both SA and Uganda having public sector HPV vaccination programs [29, 30] almost a quarter of women did not recognize that HPV infection was a risk factor for cervical cancer. Interestingly, in both SA and Uganda having many sexual partners was the most recognized cervical cancer risk factor, suggesting that although women know that sexual intercourse is linked to cervical cancer, the underlying mechanism (HPV infection) is not understood by most. Interventions to help women understand the link between this risk factor and cervical cancer might help increase HPV vaccination uptake. Encouragingly most women recognized abnormal vaginal bleeding and smelly vaginal discharge as symptoms of cervical cancer.

We found breast and cervical cancer symptom and cervical cancer risk factor awareness was lower in rural compared to urban sites in SA, but not in Uganda. The SA findings confirm those of a much smaller study conducted in SA in 2002 [36]. These differences may be related

to the high levels of inequity in SA as indicated by the Gini co-efficient of 0.6 for SA compared to a co-efficient of 0,4 for Uganda [37]. Inequity between urban and rural areas is also likely to include differences in access to evidence-based cancer information, prevention messaging and access to screening services. This is seen in the 2016 South African Demographic and Health Survey which reported differences in women accessing Pap smear screens by province as well as by urban versus non-urban residence [18]. Our study highlights the importance of investigating cancer awareness and knowledge in both urban and rural settings when developing evidence to inform and implement cancer awareness interventions. Our study also provides information on women's access to the radio, television and internet by geographic area which could inform how best to deliver awareness interventions in the different settings.

Similar to other studies we also found that in SA higher education levels were associated with greater breast cancer symptom awareness [9]. Our finding that women doing paid work in SA had lower breast cancer symptom awareness compared to those not doing paid work was surprising and further research is required to confirm this finding.

For Uganda, the socio-demographic variables that were significantly associated with breast cancer awareness were relationship status and age, with women not living with a partner/not having a partner and women younger that 30 years compared to older women reporting lower awareness of breast risk factors and symptoms respectively. A study on breast cancer awareness among women recently diagnosed with breast cancer in Uganda also showed that unmarried women had lower breast cancer awareness [8]. Similar to a study in Kinshasa we found that women not living with a partner/not having a partner in Uganda had lower cervical cancer awareness compared to women living with a partner [21]. These findings underscore the importance of cancer awareness programs reaching specific target groups.

Belief in witchcraft has been shown to be important in disease conceptualization for both communicable and non-communicable diseases, although less research has been undertaken on belief of witchcraft as cause of cancer [26, 38]. We found that few participants in Uganda and SA identified witchcraft as a risk factor for either cancer. This finding may be the result of changing beliefs, but needs to be interpreted with caution given that we only used one question to assess this belief. Witchcraft beliefs are complex and community-specific. Future studies would benefit from using multiple items to measure this complexity as well as qualitative research to better explore community- specific beliefs.

We identified important lay beliefs including: putting money in a bra and wearing a tight bra (the commonest lay beliefs held in SA and Uganda respectively about increasing risk for breast cancer); poor personal hygiene and insertion of objects, creams or herbs (the most commonly held lay beliefs about increasing risk for cervical cancer). Other African studies have reported on the belief that vaginal insertion of creams and objects, employed as a means to increase partner sexual satisfaction or as part of a hygiene practice, could be a cause of cervical cancer [39–41]. Beliefs related to wearing a tight bra have also been reported previously [9]. If individual behavior is directed by lay beliefs this could result in a false sense of protection against cancer or a misattribution of symptoms and delayed help-seeking. Therefore, it is important that intervention programs not only provide information on evidence-based risk factors and symptoms but also address lay beliefs.

A major strength of our study was the use of a locally validated questionnaire—this enabled comparison across our study sites in urban and rural locations in two countries, and will allow for accurate measurement of changes in awareness over time. Many studies measuring cancer awareness have been conducted at health facilities amongst women diagnosed with cancer, with information assumed to hold true for undiagnosed women. A strength of our study was that it was community-based with a random selection of homesteads across different locations, thereby minimizing selection bias. A further strength was that our questionnaire was available

in local languages (Acholi in Uganda and isiXhosa in SA) as well as in English, allowing for true community participation—this together with the PAC support facilitated community access and could explain our very high response rate. The use of tablets customized with the questionnaires and with off-line capability further aided efficient data collection, particularly in the rural areas with poor internet connectivity.

A limitation of our study was that our questions assessing recognition of breast and cervical cancer symptoms were slightly different from the recall questions for breast and cervical cancer symptoms. For example for breast cancer we asked specifically "Please would you name as many symptoms or signs of breast cancer as you can think of?" whereas for recognition we expanded the question to "Can you tell me if you think the following could be signs of something serious or that something is wrong, such as breast cancer?" This was done to acknowledge that symptom interpretation as something "serious" is likely to drive help-seeking rather than just attribution specifically to cancer. But it does mean we are not comparing like with like with recall (cancer symptoms only) vs recognition. The differences in levels of knowledge measured by unprompted versus prompted questioning highlight the need for careful consideration of how knowledge is measured and compared between studies. A further limitation of our study was that the development of our asset index was informed by the few socio-demographic variables we collected. It is possible that an asset index informed by more variables might have elicited an association between cancer awareness and socio-economic status as has been described in other studies [10, 42]. Addressing the two leading female cancers in SSA in one study is a resourceful way to maximise data collection from women and minimises duplication between studies. However, this did result in a long questionnaire and a possible limitation of this approach is participant fatigue.

## Conclusion

Our study identified important and specific gaps in community breast and cervical cancer risk and symptom awareness as well as cancer-related lay beliefs in rural and urban settings in SA and Uganda. These results could inform targeted intervention programs in similar urban and rural locations in both countries e.g. media campaigns to increase awareness of the non-lump symptoms for breast cancer. Our findings also provide an important baseline measure for the evaluation of future interventions.

## Supporting information

**S1 Appendix. Participant recruitment.**
(DOCX)

**S2 Appendix. Participants' assets by site, country and overall.**
(DOCX)

**S3 Appendix. Recalled and recognized breast cancer risk factors and symptoms by site.**
(DOCX)

**S4 Appendix. Modified Poisson regression showing socio-demographic predictors of higher versus lower breast cancer risk factor and symptom awareness in South Africa.**
(DOCX)

**S5 Appendix. Modified Poisson regression showing socio-demographic predictors of higher versus lower breast cancer risk factor and symptom awareness in Uganda.**
(DOCX)

**S6 Appendix. Unprompted and prompted breast and cervical cancer lay beliefs.**
(DOCX)

**S7 Appendix. Recalled and recognized cervical cancer risk factor and symptom by site.**
(DOCX)

**S8 Appendix. Modified Poisson regression showing socio-demographic predictors of higher versus lower cervical cancer risk factor and symptom awareness in South Africa.**
(DOCX)

**S9 Appendix. Modified Poisson regression showing socio-demographic predictors of higher versus lower cervical cancer risk factor and symptom awareness in Uganda.**
(DOCX)

## Acknowledgments

The authors thank the study participants for sharing their knowledge and experiences; Dr Jennifer Githaiga for her commitment, enthusiasm and support throughout the project; the project advisory committee members, field staff and community liaison managers for their support in data collection; our students for assistance in formatting tables and figures; the Outcome Registry Intervention and Operation Network (ORION) team, who developed and hosted the electronic surveys on their platform. FMW is Director and SES is co-investigator of the multi-institutional CanTest Collaborative, which is funded by Cancer Research UK (C8640/A23385).

## Author Contributions

**Conceptualization:** J. Moodley, A. D. Mwaka, S. E. Scott, F. M. Walter.

**Data curation:** D. Constant.

**Formal analysis:** J. Moodley, D. Constant, S. E. Scott.

**Funding acquisition:** J. Moodley, F. M. Walter.

**Methodology:** J. Moodley.

**Project administration:** J. Moodley, A. D. Mwaka.

**Writing – original draft:** J. Moodley.

**Writing – review & editing:** J. Moodley, D. Constant, A. D. Mwaka, S. E. Scott, F. M. Walter.

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
