## [Decision Letter · Decision Letter 0]

6 May 2020

PONE-D-20-06115

Mapping breast and cervical cancer awareness in Uganda and South Africa

PLOS ONE

Dear Prof Moodley,

Thank you for submitting your manuscript to PLOS ONE. After careful consideration, we feel that it has merit but does not fully meet PLOS ONE’s publication criteria as it currently stands. Therefore, we invite you to submit a revised version of the manuscript that addresses the points raised during the review process.

We would appreciate receiving your revised manuscript by Jun 20 2020 11:59PM. To enhance the reproducibility of your results, we recommend that if applicable you deposit your laboratory protocols in protocols.io, where a protocol can be assigned its own identifier (DOI) such that it can be cited independently in the future. For instructions see: http://journals.plos.org/plosone/s/submission-guidelines#loc-laboratory-protocols

We look forward to receiving your revised manuscript.

Kind regards,

Joel Msafiri Francis, MD, MS, PhD

Academic Editor

PLOS ONE

Journal Requirements:

Reviewers' comments:

Reviewer's Responses to Questions

**Comments to the Author**

1. Is the manuscript technically sound, and do the data support the conclusions?

Reviewer #1: Yes

Reviewer #2: Partly

2. Has the statistical analysis been performed appropriately and rigorously? 

Reviewer #1: Yes

Reviewer #2: No

3. Have the authors made all data underlying the findings in their manuscript fully available?

Reviewer #1: Yes

Reviewer #2: No

4. Is the manuscript presented in an intelligible fashion and written in standard English?

Reviewer #1: Yes

Reviewer #2: Yes

5. Review Comments to the Author

Reviewer #1: Mapping Breast and Cervical awareness in Uganda and South Africa.

1. This is a very good paper; however, I find it so ‘bulky’ to be one paper. Breast cancer awareness alone would be a good paper and so would cervical cancer awareness. In the future, consider splitting these as two separate papers.

2. The comparison between South Africa and Uganda is a good one, a low and middle-income country study is a bonus. It brings out unique findings.

3. Please mention which particular districts or region where your study was conducted in Uganda and South Africa. You mention that participants spoke Acholi and KiXhosa, but not the study location.

4. What was the recall period for the risk factors? Where did they know most about the risk factors? If this

5. I wonder why the authors chose to use chi square tests and not do further regression analysis to determine which the strongest predictors of awareness are. Chi square tests show that there is an association but it is important to show the strength of association and eliminate confounders

Reviewer #2: The manuscript titled Mapping breast and cervical cancer awareness in Uganda and South Africa aim to measure breast cancer and cervical cancer symptom and risk factor awareness and recognition and lay beliefs among urban and rural population in South African and Uganda. This is an important study, particularly in the context of the high breast and cervical burden in SSA where community awareness has not been assessed. However, I have concerns regarding the readability of the manuscript.

The title does not incorporate what is being assessed in the study. It doesn’t specify that breast and cervical cancer risk factor and symptom awareness and lay beliefs are being measured and associated socio-demographic factors assessed.

Study outcomes are not well defined. There is limited description of how scores were organised and assessed for the analysis in the materials and methods, making it difficult to interpret results specifically for the multivariate analysis. Additionally, results are not well presented making it difficult to understand whether as a reviewer or as a reader. I am also concerned that not all results, most importantly multivariable results, are presented in tables included in the manuscript. The sample selection diagram to be revised and highlight/denote whether a household represents an individual. Additionally, the reported number of households assessed in text does not match with what is reported in the diagram.

The discussion needs to be considerably strengthened. Authors should try limiting repetition of results and focus more on discussing what the findings mean and relating it back to the aim of the study, how results compare to findings from previous studies or results from national programs and providing key recommendations.

There are several grammatical errors that the authors should correct. The authors should also cite appropriately in text and references should follow journal format.

Overall, substantial revisions are required before the manuscript can be published. The authors need to be clear what the main aim of this study is and be concise about documenting study methods, reporting results and discussing the results.

6. PLOS authors have the option to publish the peer review history of their article (what does this mean?). If published, this will include your full peer review and any attached files.

Reviewer #1: Yes: JULIET NABIRYE

Reviewer #2: No

---

## [Author Response · Author response to Decision Letter 0]

16 Jun 2020

Response to reviewers : 16th June 2020 

We sincerely thank the reviewers for their insightful comments which we believe have improved the quality of our paper. 

Reviewer #1: Mapping Breast and Cervical awareness in Uganda and South Africa.

1. This is a very good paper; however, I find it so ‘bulky’ to be one paper. Breast cancer awareness alone would be a good paper and so would cervical cancer awareness. In the future, consider splitting these as two separate papers.

Thank you for the suggestion. For this paper we felt it important to present the key findings together. For future papers based on this study we will consider splitting breast and cervical cancer results into separate papers to improve readability.

2. The comparison between South Africa and Uganda is a good one, a low and middle-income country study is a bonus. It brings out unique findings.

Thank you

3. Please mention which particular districts or region where your study was conducted in Uganda and South Africa. You mention that participants spoke Acholi and KiXhosa, but not the study location.

We have intentionally not named the specific sites as this can result in stigmatisation at a community level. We have however added regional site.

“In Uganda both urban and rural sites were in Northern Uganda. In SA our urban site was in the Western Cape Province and our rural site in the Eastern Cape Province.”

4. What was the recall period for the risk factors? Where did they know most about the risk factors? If this

We did not have any specific recall period. The recall rather refers to an open-ended question.

5. I wonder why the authors chose to use chi square tests and not do further regression analysis to determine which the strongest predictors of awareness are. Chi square tests show that there is an association but it is important to show the strength of association and eliminate confounders

Results of the regression analysis are included in the text and appendices.

Reviewer 2

General comments:

Reviewer #2: The manuscript titled Mapping breast and cervical cancer awareness in Uganda and South Africa aim to measure breast cancer and cervical cancer symptom and risk factor awareness and recognition and lay beliefs among urban and rural population in South African and Uganda. This is an important study, particularly in the context of the high breast and cervical burden in SSA where community awareness has not been assessed. However, I have concerns regarding the readability of the manuscript.

The title does not incorporate what is being assessed in the study. It doesn’t specify that breast and cervical cancer risk factor and symptom awareness and lay beliefs are being measured and associated socio-demographic factors assessed.

We have revised the title to “Mapping awareness of breast and cervical cancer risk factors, symptoms and lay beliefs in Uganda and South Africa” 

Study outcomes are not well defined. There is limited description of how scores were organised and assessed for the analysis in the materials and methods, making it difficult to interpret results specifically for the multivariate analysis. 

We have added details to the methods as suggested by the reviewer.

Additionally, results are not well presented making it difficult to understand whether as a reviewer or as a reader. I am also concerned that not all results, most importantly multivariable results, are presented in tables included in the manuscript. The sample selection diagram to be revised and highlight/denote whether a household represents an individual. Additionally, the reported number of households assessed in text does not match with what is reported in the diagram.

We have revised results in line with the reviewer’s comments

The discussion needs to be considerably strengthened. Authors should try limiting repetition of results and focus more on discussing what the findings mean and relating it back to the aim of the study, how results compare to findings from previous studies or The results from national programs and providing key recommendations.

The discussion has been revised

There are several grammatical errors that the authors should correct. The authors should also cite appropriately in text and references should follow journal format.

Grammatical errors have been corrected. Citations have been added.

Abstract

The tools should not be included in the background, it can be incorporated in the methods section as follows: “Data were collected by interviewers using electronic tablets customised with the locally validated African Women Awareness of Cancer (AWCAN) tool.”

We have deleted this from the background and revised methods to read “Data were collected by interviewers using electronic tablets customised with the locally validated African Women Awareness of Cancer (AWCAN) tool.”

It would be helpful to mention test statistics used in your analysis.

Revised as follows “Mann Whitney and Kruskal Wallis tests were used to compare the association between sociodemographic variables and outcomes. Poisson regression with robust variance was conducted to identify independent socio-demographic predictors.” 

Authors to report percentages up to one decimal point (i.e. 90.8% vs 91% for cervical cancer awareness)

We have revised - all percentages are now reported up to one decimal point.

I think findings serve as foundation for further studies and development of interventions.

The last sentence in the abstract has been revised to read:

“Our results serve as a foundation for further studies and development of interventions.”

Background:

line 73, “the” is missing, so the sentence should be “Breast and cervical cancer are the leading causes…”

This has been revised

Line 76-80, needs a reference 

The data is from Globocan we have added the reference

Line 80, Please write out in full before using abbreviations (UK and SA)

These abbreviations were written in full (previously lines 85 and 86, now lines 80 and 81)

Line 80 - 82, This doesn’t read well. I understand that they present with late stage disease, but do you mean to convey that the main motivator for presenting for care are symptoms? if so please review this sentence so that this is communicated clearly.

Revised to “Most SSA countries do not have cervical or breast cancer screening programs and the majority of cancers are diagnosed symptomatically and at an advanced stage.”

Line 82-83, Please also include a reference for low survival rates for cervical cancer current reference included only relates to breast cancer survival rates.

We have added a reference for cervical cancer.

Line 82 - 83, Perhaps “Late-stage at presentation is a major contributor to low survival rates” should be “Late-stage breast and cervical cancer at presentation is a major contributor to low survival rates”

Revised to “Late-stage at presentation is a major contributor to low breast and cervical cancer survival rates”

Line 85 -86, This is not well-structured sentence. The independent clause should either come at the beginning or end of your compound complex sentence. I suggest you amend it as follows: “Early stage presentation is a key goal of comprehensive cancer policies as it enables more opportunities for curative treatment and improved prognosis a month those treated.”. Also, include a reference for this statement.

We have revised to “Early stage presentation is a key goal of comprehensive cancer policies as it enables more opportunities for curative treatment and improved prognosis among those treated” and have added a reference.

Line 86 -88, “Prompt help-seeking in the presence of symptoms…”. Help seeking should not only be motivated by symptoms. Prevention efforts should really be also addressing risk perception and perceived benefits for accessing screening.

We agree that screening is an important component of early cancer diagnosis, however the focus of our paper is on symptomatic cancer.

Line 92, It should be “…nature of…” and “predictors” (plural). Additionally, it is simpler to just say “Understanding awareness of cancer symptom and risk factor is vital to underpin the development of locally relevant interventions”

We have reworded the sentence as follows “Understanding the nature and predictors of cancer symptom and risk factor awareness is vital to underpin the development of locally relevant interventions”

Line 94, I assume “lay workers” should be “lay health workers”

We have change to” lay health workers”

Lin 95 -101, It will be useful to helpful to re current estimates present current breast/cancer awareness in Sub-Saharan Africa and specifically in the countries being studied (Uganda and South Africa). I know you mention that current estimates are mostly from hospital-based study populations, but even that can serve as starting point.

The lack of validated cancer awareness measurement tools has prevented us knowing this and makes comparison of previous surveys difficult. We have revised the sentence in the paragraph that follows to make this point (lines 105-106)

“Furthermore, most studies in SSA, whether hospital or community-based have not used locally validated measures.”

Line 103, I suggest changing “be very different” to “differ”

We have made this change as suggested by the reviewer.

Line 104 -108, This should be part of Materials and method section. Instead, authors should put in the aim of the study here, before moving on to materials and methods. 

We have moved this part to the Materials and methods. The paragraph now ends with our aim.

“The aim of this study was to measure breast and cervical cancer symptom and risk factor awareness and lay beliefs in urban and rural settings in Uganda (a low-income country) and SA (a middle-income country) using a culturally relevant, locally validated measurement tool” 

Materials and methods

Line 113, “18 years and over” should be “18 years and older”

Revised to “18 years and older”

Line 117, There’s a comma missing after “AWACAN project”

Comma inserted.

Line 121, There’s a comma missing after “In this study”

Comma inserted.

Line 127-128, Please provide more details regarding the sampling approach including randomisations procedures. What if a household selected does not have a woman, or the woman fused? Also, what if the first woman approached has cancer history but there is another woman in the same household that doesn’t have a history of cancer? 

We have added details on the sampling approach. The paragraph now read “At each site, households were selected using systematic random sampling. The name of each woman in the household was written on a separate piece of paper, folded and placed in a “hat”. Random selection of a participant was then done by picking one from the set of folded papers. If the woman selected did not want to participate or there were no women in the household, fieldworkers moved to the next marked household, noting the reason for non-participation. If the selected woman from the household was away, fieldworkers returned for an interview later. Women unable to speak either Acholi, isiXhosa or English; women with a history of breast or cervical cancer and; woman younger than 18 years were excluded.”

Line 135 - 136, Were the interview done in private, as this can affect participant responses?

Interviews were conducted in a private space. We have revised the sentence as follows “Women were interviewed face-to-face in a private space in their homes, either in their local language (isiXhosa or Acholi) or in English depending on their preference “

Line 140 – 141, I assume “education level” should be “highest education level”

Thank you we have changed to “highest education level”

Line 141, Employment status is called “paid work” in the tables – please use consistently in text and tables and ensure that it's defined accordingly in the text.

We have revised to “paid work” throughout.

Line 143, Asset index should not be capitalised as it’s not a proper noun, please apply this consistently throughout text

We have changed throughout the text.

Line 154 – 155, Please provide a brief provide description of questions asked to assess cancer awareness, similar to the approach used to describe knowledge of risk factors and symptoms in the subsequent sentence. Also, please provide a reference to the AWACAN breast and cervical cancer tool. 

We have added a brief description of the questions used to assess cancer awareness and added the reference for the AWACAN tool.

We measured cancer awareness by asking whether the participant had heard of breast/cervical cancer and whether they knew of any family members, friends or neighbours who had breast/cervical cancer. Risk factor and symptom awareness was assessed using an open/unprompted question followed by closed/prompted question format as laid out in the AWACAN breast and cervical cancer tool [15]

Line 157, “women’s” should be “woman’s” the “any” denotes singular, but women is plural

We have revised.

Line 158, “names” should be “name” it is a verb in this sentence 

This has been revised.

It would also be helpful to reference the relevant appendices for each item describe (i.e. breast cancer signs and symptoms (S3 appendix)

We have referenced the AWACAN tool which has the specific items. 

Line 168 – 172, More details are needed here regarding how the authors handled the scores and how they were interpreted, i.e. did higher scores indicate better/higher recognition of signs and symptoms? Also, how will these scores be reported and analysed, as continuous variable (averages/means/ medians) or categorize, (what cut-off are used)?

We have added more detail.

Higher scores indicated better recognition of symptoms and risk factors and greater endorsement of lay beliefs. The median and the interquartile range (IQR) are reported for symptom and risk factor scores. Similar to other studies we dichotomised scores for risk factor and symptom awareness using the median as a cut-off for low and high awareness [18-20]. The median was classified as low awareness to achieve an even distribution between groups.

Line 178 – 179, was the consent form administered in the participants’ own/preferred language, and were consent forms translated?

We have added the following “Consent forms were translated and administered in the participants’ preferred language.”

Line 189, “Sociodemographic” does not have a hyphen which is inconsistent to previous format in the manuscript.

We have changed to socio-demographic throughout.

Line 191, The abbreviation IQR should come after ranges as it encompasses the word “range”

This has been revised.

Line 192, up until now South Africa has been abbreviated to SA. Please choose format (abbreviated or written in full) and use consistently throughout the manuscript.

Revised so that it is written in full the first time and abbreviated thereafter.

Line 195, “Sociodemographic” does not have a hyphen which is inconsistent to previous format in the manuscript.

Changed to socio-demographic throughout.

Line 196 – 197, If I understand correctly, the authors dichotomised scores for awareness using the median as a cut-off for low and high based? Firstly, this should be described in detail in the “Measures of cancer awareness” section. Secondly, the authors should substantiate why they used this cut-off, and provide reference to support this, if any?

Revised and moved to “Measures of cancer awareness” section “Higher scores indicated better recognition of symptoms and risk factors and greater endorsement of lay beliefs. The median and the interquartile range (IQR) are reported for symptom and risk factor scores. Similar to other studies we dichotomised scores for risk factor and symptom awareness using the median as a cut-off for low and high awareness [18-20]. The median was classified as low awareness to achieve an even distribution between groups.”

Line 194 -199 Logistic regression is not appropriate to use in this instance. Odd ratio has been shown to overestimate associations if the outcomes is not rare (i.e. above >=10% prevalence). I suggest using log binomial or Modified Poisson regression analysis to estimate relative risk. See below a reference that talks to this:

Lovasi, G. S., Underhill, L. J., Jack, D., Richards, C., Weiss, C., & Rundle, A. (2012). At odds: concerns raised by using odds ratios for continuous or common dichotomous outcomes in research on physical activity and obesity. The open epidemiology journal, 5, 13.

We have revised used Poisson regression with robust variance to identify independent socio-demographic predictors. 

Results

S1 Appendix,

The total household visited do not add up to 1941, they add up wot 1940 if you add up all the county totals

Thank you for pointing this out. The error was with the Ugandan rural household visited which should be 443. The total household visits is 1941.

Does a household represent an individual? This needs to be clarified in the methods and diagram.

We have clarified in the methods and have revised the diagram as follows: households visited, individuals refused, individuals ineligible, enrolled individual

Line 204, The overall refusal rate reported in text does not match what is reported in S1 Appendix

We have corrected this to 6.3%

Line 205, Refusal rate for urban Uganda sites should be 13.7%

Refusal rate now reported to one decimal.

Line 209, Authors to include median age and IQR in Table 1. 

Table revised and information added. 

Table 1, 

Categories for age should be labelled 18-29, 30-50, ≥50. 

We have revised to 18-29, 30-49, ≥50 

Current Table 1 format is unnecessarily expansive. It should be adjusted to include number, with percentage in brackets in one column. (same applies to S1-S5 Appendices)

We have revised with number (percentage) in one column

It’s not necessary to include a total sample size for each variable. Instead. The sample size should only be included in the column titles for each country (i.e. total for each country, then totals for urban and rural area for each country). Revised as recommended 

Information on missing data to be stated (if any). Information added

I find it odd that there no participants in a relationship but not living together, can the authors confirm this finding? 

We acknowledge that the category “single” is confusing. We included in this category anyone who either did not have a partner or was not living together with a partner. We have revised the category labels to: Married/living with partner; No partner/ Not living with partner; Widowed/Separated/Divorced.

Line 211, Authors to report percentages up to one decimal point (i.e. 66.3% vs 66% for Paid work)

Revised to report up to one decimal point.

Line 221, Knowing of someone with the disease is not reported anywhere in the tables, and this was not defined in the materials and methods section.

We have now added this to the methods (lines 240) and have elected to report the results in the text only.

Line 224, These results (9% of 1596) are not presented anywhere on the tables or appendices. There’s no total column in S3 appendix where I assume these results would be listed. The overall summary of these finding should also be included in one of the tables in the manuscript.

The appendix has been revised to include these findings and a total column.

Line 224, These results (95%) are not presented anywhere on the tables or appendices. There’s no total column in S3 appendix where I assume these results would be listed. The overall summary of these finding should also be included in one of the tables in the manuscript.

The appendix has been revised to include these findings and a total column.

Line 229 – 231, These results (overall commonly recognized and least recognized risk factors) are not presented anywhere on the tables or appendices. There’s no total column in S3 appendix where I assume these results would be listed.

The appendix has been revised to include these findings and a total column.

Line 234, The figure included is not very clear. The risk factors are blurry and it’s difficult to read. 

The figures were converted to TIFF files using the PACE tool as recommended by PloSOne. We have now changed some of the column colors and text color and the figure meets the journal requirements.

Line 240 – 243, These results (multivariable analyses) are not presented anywhere on the tables.

We have added the Poisson regression results as appendices S4 and S5.

Line 248 -252, These results (overall breast cancer symptom and recognition) are not presented anywhere on the tables or appendices. There’s no total column in S3 appendix where I assume these results would be listed.

The appendix has been revised to include these findings and a total column.

Line 255, “Education” should be “highest education level”. Employment should be “paid work”

We have revised accordingly.

 Line 256 -259, These results (multivariable analyses) are not presented anywhere on the tables or appendices. 

We have added the Poisson regression results as appendices S4 and S5.

Line 260 -262, These results (multivariable analyses) are not presented anywhere on the tables or appendices.

We have added the Poisson regression results as appendices S4 and S5.

Line 265, 20% should be 21.0% revised to 21.1%

Line 267, 98% should be 97.8% revised to 97.8%

Line 268, include the following statement “…in prompted questions” before the bracket 

We have added “in prompted questions” before the bracket.

Line 265, 1% should 0.8% and 22% should be 22.2% 

We have revised and reported all percentages to one decimal point.

Line 274, These results (38% of 1571) are not presented anywhere on the tables or appendices. There’s no total column in S5 appendix where I assume these results would be listed. 

The appendix (now S7) has been revised to include these findings and a total column.

Line 276, Having unprotected sex is the most commonly recalled risk factor in urban SA. In Rural SA, it is having other sexually transmitted diseases. 

Revised to “In Uganda, having multiple sexual partners was the most commonly recalled risk factor. In urban SA having unprotected sex was the most commonly recalled risk factor, whilst in rural SA having other sexually transmitted diseases was most commonly recalled.”

Line 277 - 278, These results (<1%) are not presented anywhere on the tables or appendices. There’s no total column in S5 appendix where I assume these results would be listed. . 

The appendix (now S7)has been revised to include these findings and a total column.

Line 281 - 284, These results are not presented anywhere on the tables or appendices. There’s no total column in S5 appendix where I assume these results would be listed. . 

The appendix (now S7) has been revised to include these findings and a total column.

Line 286 – 287, 8% should be 7.9%, 27% should be 26.6%, 38% should be 37.8%, and 31% should be 30.7% 

We have revised and reported all percentages to one decimal point.

Line 289, The figure included is not very clear. The risk factors are blurry and it’s difficult to read.

The figures were converted to TIFF files using the PACE tool as recommended by PloSOne. We have now changed some of the column colors and text color and the figure

Line 292 – 294, These results (multivariable analyses) are not presented anywhere on the tables. 

We have added the Poisson regression results as appendices S8 and S9.

Line 296 (page 18), These results (multivariable analyses) are not presented anywhere on the tables. 

We have added the Poisson regression results as appendices S8 and S9.

Line 297, These results are not presented anywhere on the tables or appendices. There’s no total column in S5 appendix where I assume these results would be listed. 

The appendix (now S7) has been revised to include these findings and a total column.

Line 301, The figure included is not very clear. The risk factors are blurry and it’s difficult to read. 

The figures were converted to TIFF files using the PACE tool as recommended by PloSOne. We have now changed some of the column colors and text color and the figure

Line 304 – 310, These results (multivariable analyses) are not presented anywhere on the tables. 

We have added the Poisson regression results as appendices S8 and S9.

Line 314, 93% should be 92.3%, and 34% should be 33.7% Revised

Line 316, 85% should be 85.4% Revised

Line 317 – 318, <1% should be overall <0.5%, and 22% should be 22.3% 

Revised

Discussions

Line 328, “In keeping with other studies…) should be “In line with findings from previous studies, ) 

Revised to “In line with findings from previous studies,…”

Line 329, “…symptom awareness was higher with prompted questioning” 

Revised as suggested.

Line 332, This paragraph should be summarised in the limitations. We have summarised in the limitations.

Line 344 - , should move to line 328, after the first paragraph of the discussion 

We have moved to after the first paragraph.

Line 347, better to say responsive rather than amenable. Also, can you include reference that support the statement regarding exercise as an effective intervention for being overweight. 

We have changed “amenable” to responsive. 

The statement “Similar to other studies in low-and middle-income African settings, we found poor levels of knowledge for factors which are responsive to interventions such as physical exercise and being overweight.” provides examples of two of the risk factors responsive to interventions rather than indicating that exercise is an effective intervention for being overweight. 

Line 348, The authors should include reference for breast cancer risk factors being common to other NCDs. Also, it is generally more cost effective to adapted existing interventions to include additional messaging. The authors should maybe highlight this seeing as there are risk factors common to other NCDs. 

We have added a reference for breast cancer risk factors being common to other NCDs and have added a sentence about the cost-effectiveness of adapting existing interventions

“Adapting existing interventions to include additional messaging will be an efficient and cost-effective approach to addressing low breast cancer risk factor awareness.”

Line 355, There should be a common after the reference not a full stop. 

Revised

Line 362 - 364, References after need here, including the respective guidelines. 

We have added relevant references.

Line 367 – 368, References needed for the statement that screening coverage is suboptimal is SSA. We have added a reference.

Line 369, There should not be a comma after screening. Also, authors should include the SA cancer prevention guidelines as a reference here in addition to the current reference. 

Comma deleted and SA cancer prevention guidelines added as a reference.

Line 373, What do the authors mean by opportunistic screening programs? This needs to be clarified, and the reasons why it seems to mostly be limited to national and regional referral hospitals. 

Opportunistic screening in Uganda occurs when individuals who have visited the health facilities for other reasons are offered a cervical screening test. There are no reminder systems for screening and for receiving results of the screening tests in Uganda. 

We have revised this to read “Limited opportunistic screening takes place at national referral and regional referral hospitals when women attend for other reasons. Studies have reported low self-reported cervical screening uptake among community women in different parts of Uganda, ranging from 4.8% to 7.0% [32,33].”

Remove the bracket after 7%. Also, 7% should be 7.0% 

Revised

Line 375, A reference is needed for program screening HIV infected women more frequently. We have added references.

Line 376, I would rather say findings are unexpected, not disappointing. How do these results compare to previous results, or reported results from national programs? Also, Authors should explore the reasons why HIV was not well recognised as a risk factor for HIV among the study population. Is there data on HIV prevalence amongst your study participants? If not, I’m sure that there are HIV estimates for communities involved in the study? Seeing as SSA has highest burden of HIV, one would expect that exposure to HIV service would lead to some exposure to cervical cancer prevention messaging seeing as HIV infected women are at higher risk. Additionally, cervical cancer prevention and screening guidelines are integrated into HIV treatment guidelines in SA (since 2010), as should screening and treatment services for HIV infected women. Granted access to integrated services may be limited in rural areas, but it still doesn’t explain the low recognition of HIV as a risk factor for cervical cancer.

We did not collect data on HIV status of our participants, but have added country level HIV prevalence data to the discussion. Only a few studies have assessed women’s awareness of HIV as a risk factor for cervical cancer in our settings. We have added information from these studies to allow for some comparison. The revised paragraph reads as follows:

“Eastern and Southern Africa are the regions most affected by HIV worldwide. The HIV prevalence among women aged 15 to 49 years is 25.8% in South Africa and 7.1% in Uganda [34,35] The association between HIV and cervical cancer is well established [36,37] and most screening programs offer more frequent screening for HIV infected women [26,38] In SA cervical cancer screening guidelines have been incorporated into HIV services for HIV infected women [26]. Despite these efforts, almost half the women in our study did not recognize that HIV was a risk factor for cervical cancer. Only a few studies in Africa have assessed women’s awareness of HIV as a risk factor for cervical cancer. A cross-sectional study conducted among HIV infected women attending an urban clinic in West Africa reported that only 42.5% of women identified HIV as a risk factor for cervical cancer [39]. Okunowo et al reported even lower awareness (20.5 %) among women attending a gynaecology outpatient clinic at a tertiary hospital in Nigeria [40]. In contrast to these findings, a community-based study in Eastern Uganda reported that 74.7% of participants were aware that HIV was a risk factor for cervical cancer [41]. Reasons for the much higher awareness reported in Eastern Uganda need to be explored as they could provide insights on how to address risk factor awareness in other settings. Efforts to improve community awareness around the benefits of screening and about the link between HIV and cervical cancer may help increase screening coverage and reduce the prevailing high cervical cancer incidence.”

Line 383, Authors to include references for SA and Uganda HPV vaccine programs. 

References added

Line 381 – 389, Again, authors need to discuss how these results compare to previous studies or reported results from national programs. 

We have revised and included results from a systematic review of HPV knowledge.

“A systematic review of HPV vaccine acceptability studies conducted in Africa found that although vaccine acceptability was high, HPV knowledge was low [42]. Across nine studies, less than a third of participants knew of a relationship between HPV infection and cervical cancer. Subsequent to this review HPV vaccine program have been introduce in SA and Uganda [26,27]. In our study we found higher awareness of HPV as a risk factor for cervical cancer than reported in the review article, but still almost a quarter of women did not recognize that getting an HPV infection was a risk factor for cervical cancer. Interestingly, in both SA and Uganda having many sexual partners was the most recognized cervical cancer risk factor, suggesting that although women know that sexual intercourse is linked to cervical cancer, the underlying mechanism (HPV infection) is not understood by most. Interventions to help women understand the link between this risk factor and cervical cancer might help increase HPV vaccination uptake.”

Line 391 – 397, Again, authors need to discuss how these results compare to previous studies or reported results from national programs. The Gini coefficient explains the unequal distribution of wealth between communities, but more needs to be said regarding how this may lead to unequal health care service access, including prevention messaging and screening services which may differ in these circumstances. 

We have revised this paragraph to include results from a study comparing urban and rural cancer awareness and to add more about the inequity in SA.

We found breast and cervical cancer symptom and cervical cancer risk factor awareness was lower in rural compared to urban sites in SA, but not in Uganda. The SA findings confirm those of a much smaller study conducted in SA in 2002 [43]. These differences may be related to the high levels of inequity in SA as indicated by the Gini co-efficient of 0.6 for SA compared to a co-efficient of 0,4 for Uganda [44]. Inequity between urban and rural areas is also likely to include differences in access to evidence-based cancer information, prevention messaging and access to screening services. This is seen in the 2016 South African Demographic and Health Survey which reported differences in women accessing Pap smear screens by province as well as by urban versus non-urban residence [45]. Our study highlights the importance of investigating cancer awareness and knowledge in both urban and rural settings when developing evidence to inform and implement cancer awareness interventions. Our study also provides information on women’s access to the radio, television and internet by geographic area which could inform how best to deliver awareness interventions in the different settings. 

Line 413 – 415, Additionally, witchcraft related beliefs are community specific, and this also need to be considered in future research related to this topic. We have revised to read

Witchcraft beliefs are complex and community-specific. Future studies would benefit from using multiple items to measure this complexity as well as qualitative research to better explore community-specific beliefs.

Line 416 – 419, There’s been previous studies related to insertion of objects for example in HIV prevention studies (related to condom use/non-use). Though this doesn’t relate to cervical cancer directly, it may provide some information on issues related to health risk behaviour among people that believe in such practices which authors may compare and contrast to findings in the current study. 

We have added findings from other studies on vaginal practices and beliefs.

“Other African studies have reported on the belief that vaginal insertion of creams and objects, employed as a means to increase partner sexual satisfaction or as part of a hygiene practice, could be a cause of cervical cancer [47-49]. Beliefs related to wearing a tight bra have also been reported previously [9]. If individual behavior is directed by lay beliefs this could result in a false sense of protection against cancer or a misattribution of symptoms and delayed help-seeking. Therefore, it is important that intervention programs not only provide information on evidence-based risk factors and symptoms but also address lay beliefs. 

Line 424 – 438, This needs to be more concise, there’re repetitions. 

We have revised as follows:

“A major strength of our study was the use of a locally validated questionnaire - this enabled comparison across our study sites in urban and rural locations in two countries, and will allow for accurate measurement of changes in awareness over time. Many studies measuring cancer awareness have been conducted at health facilities amongst women diagnosed with cancer, with information assumed to hold true for undiagnosed women. A strength of our study was that it was community-based with a random selection of homesteads across different locations, thereby minimizing selection bias. A further strength was that our questionnaire was available in local languages (Acholi in Uganda and isiXhosa in SA) as well as in English, allowing for true community participation - this together with the PAC support facilitated community access and could explain our very high response rate. The use of tablets customized with the questionnaires and with off-line capability further aided efficient data collection, particularly in the rural areas with poor internet connectivity. 

Line 440 – 446, Include the issue regarding prompted vs. unprompted questioning as a limitation. 

We have now included this under limitations.

---

## [Decision Letter · Decision Letter 1]

14 Aug 2020

PONE-D-20-06115R1

Mapping awareness of breast and cervical cancer risk factors, symptoms and lay beliefs in Uganda and South Africa

PLOS ONE

Dear Dr. Moodley,

Thank you for submitting your manuscript to PLOS ONE. After careful consideration, we feel that it has merit but does not fully meet PLOS ONE’s publication criteria as it currently stands. Therefore, we invite you to submit a revised version of the manuscript that addresses the points raised during the review process.

We look forward to receiving your revised manuscript.

Kind regards,

Joel Msafiri Francis, MD, MS, PhD

Academic Editor

PLOS ONE

Reviewers' comments:

Reviewer's Responses to Questions

**Comments to the Author**

1. If the authors have adequately addressed your comments raised in a previous round of review and you feel that this manuscript is now acceptable for publication, you may indicate that here to bypass the “Comments to the Author” section, enter your conflict of interest statement in the “Confidential to Editor” section, and submit your "Accept" recommendation.

Reviewer #3: All comments have been addressed

Reviewer #4: (No Response)

2. Is the manuscript technically sound, and do the data support the conclusions?

Reviewer #3: Partly

Reviewer #4: Yes

3. Has the statistical analysis been performed appropriately and rigorously? 

Reviewer #3: Yes

Reviewer #4: Yes

4. Have the authors made all data underlying the findings in their manuscript fully available?

Reviewer #3: Yes

Reviewer #4: Yes

5. Is the manuscript presented in an intelligible fashion and written in standard English?

Reviewer #3: Yes

Reviewer #4: Yes

6. Review Comments to the Author

Reviewer #3: Dear Author

This is a very good attempt to map the awareness level of two countries in relation to two important cancers among women. Content of this manuscript is very general and there is no new things to understand or practice. Your work is very lengthy and it is very difficult to read and understand this manuscript with so much content with many tables. I appreciate your efforts but thee is nothing to learn from this scientific piece. I took many hours to read it and understand before reaching to a conclusion.

Reviewer #4: The amount of results presented is too much and appears to be beyond the scope of a single manuscript. While the authors have made sincere efforts to bring all the content in one manuscript, the number of variables(awareness of risk factors, symptoms and beliefs for two cancers) studied and the number of settings compared(urban and rural areas of two countries) makes it complicated. It would be worth seriously considering to split the manuscript into two in order to give justice to the amount of efforts and resources invested.

68, 69 – The conclusion statement is too generic. The same words can be written for almost any descriptive study. It needs to be made a bit more specific.

114-115 - “We conducted a community-based cross-sectional survey of women aged 18 years and older

in one urban and one rural site in Uganda and SA (total of 4 sites).” How were the sites selected and how much were they representative of the respective countries in terms of socio-economic and demographic variables. Also a brief description of the two countries in terms of number of states, population, etc. may help. The paper seems to generalize the findings for the two countries respectively from the study, which may not be possible if there are widespread differences across different states within each country.

174 -175 - “We used the same open and closed questions to measure recall and recognition of cervical cancer

risk factors (11 items) and symptoms (11 items)” - the meaning conveyed from this line is not clear. One of the limitation in the discussion states that the questions for recall and recognition were asked differently.

206-207 - “Medians and interquartile ranges (IQR) are reported for continuous

variables.” - to be written in past tense

Tables 2,3,4 and 5 – Please mention the test in the footnotes. Based on my interpretation from the description, the places where z value is reported is meant for the Man Whitney U test. Those who have not heard of breast cancer should not be included in symptom identification as the awareness about the symptom if the person is not aware about the disease becomes irrelevant.

In methodology there is mention of only prompted questions on lay beliefs whereas the results discuss both prompted and unprompted ones. For measuring lay beliefs, unprompted questions may be considered more appropriate. As for the prompted questions, the respondent may not be holding the belief but may choose one from the responses which may be biased(eg - social desirability bias)

337 Table 4 has a hastag – the footnote is missing

391 “Adapting

existing interventions to include additional messaging will be an efficient and cost-effective

approach to addressing low breast cancer risk factor awareness.” - This statement should be supported with appropriate evidence.

406-407 - “Of concern is the finding that

many women in rural SA and urban and rural Uganda, did not identify lack of screening as a

risk factor for cervical cancer ??” Considering a lack of screening as a risk factor for cervical cancer may not be appropriate as most other risk factors are related to the occurrence of cervical cancer while lack of screening may be a risk factor for delayed presentation of cervical cancer but technically it may be in-appropriate to consider it as a risk factor for occurrence of cervical cancer.

408 -”Poor screening awareness has also been reported elsewhere in

Africa [28,29]. Cervical cancer screening coverage is suboptimal in SSA [30]. In SA, after

409 more than 15 years of a public sector screening program national screening coverage is at

65%, with some provinces having coverage rates as low as 46% [26,31]. In Uganda, there is

no national population based cervical cancer screening program. Limited opportunistic

screening takes place at national referral and regional referral hospitals when women attend

for other reasons. Studies have reported low self-reported cervical screening uptake among community women in different parts of Uganda, ranging from 4.8% to 7.0% [32,33].” - The description of status of screening services in the two countries may be reduced significantly or removed completely, it is not enriching the discussion in the context of the study's findings. Similarly the discussion on HIV and HPV vaccine needs to be reduced significantly.

512 – The way in which the questions related to recognition of breast and cervical cancer symptoms were asked appear to be a major limitation of the study. While the methodology described in the study talks about the measurement of awareness of breast and cervical cancer symptoms as one of the major objectives, the way in which the question has been framed “Can you tell me if you think the following could be

signs of something serious or that something is wrong, such as breast cancer?” the interpretation and responses would not have been as per the objective. This significantly reduces the confidence of the reader on the findings on the awareness of the symptoms. An interaction with the data collection team to understand how the question was interpreted by most of the respondents may help in addressing this issue.

534 – The conclusion should not be generalised for the entire countries unless the authors are extremely confident of the studied sample being representative of the entire countries. Also a few examples of how exactly can the findings can inform targeted interventions can be of great help to make the description more specific.

7. PLOS authors have the option to publish the peer review history of their article (what does this mean?). If published, this will include your full peer review and any attached files.

Reviewer #3: **Yes: **Abhishek Shankar

Reviewer #4: No

---

## [Author Response · Author response to Decision Letter 1]

24 Aug 2020

Dear Prof Francis, 

We appreciate the reviewers’ time and efforts to help improve this manuscript. We have addressed all the points provided by the reviewers and made the minor revisions to the manuscript as suggested. Please see below point-by-point responses to the reviewers’ comments. Following these minor revisions, we hope the manuscript is now acceptable for publication in PLOS ONE and look forward to hearing from you.

Response to reviewer comments 

Reviewer #3: Dear Author

This is a very good attempt to map the awareness level of two countries in relation to two important cancers among women. Content of this manuscript is very general and there is no new things to understand or practice. Your work is very lengthy and it is very difficult to read and understand this manuscript with so much content with many tables. I appreciate your efforts but there is nothing to learn from this scientific piece. I took many hours to read it and understand before reaching to a conclusion.

Reviewer #4: The amount of results presented is too much and appears to be beyond the scope of a single manuscript. While the authors have made sincere efforts to bring all the content in one manuscript, the number of variables(awareness of risk factors, symptoms and beliefs for two cancers) studied and the number of settings compared(urban and rural areas of two countries) makes it complicated. It would be worth seriously considering to split the manuscript into two in order to give justice to the amount of efforts and resources invested.

The current manuscript is fuller than the original submission due to the amendments stipulated by Reviewer 2. We acknowledge that splitting the paper may make it simpler, yet if the paper was to be split into two it would have to divide the work by country (Uganda/South Africa) or cancer type (cervical/breast) both of which would lose what has been gained by conducting an international collaboration and studying two cancer types within one study to allow comparisons. 

We therefore feel that splitting this manuscript into 2 papers would not do this work justice. We acknowledge there is a lot of data in the results and have now organized the results into 3 sections to aid the readability of the text: Section 1 outlines the Participant profile; Section 2 presents findings on awareness of breast cancer risk factors, symptoms and lay beliefs; Section 3 presents findings on awareness of cervical cancer risk factors, symptoms and lay beliefs. We are also happy for any of the tables to be added as supplementary files at the Editor’s discretion.

68, 69 – The conclusion statement is too generic. The same words can be written for almost any descriptive study. It needs to be made a bit more specific.

We have changed the conclusion to read:

“We identified gaps in breast and cervical cancer symptom and risk factor awareness. Our results provide direction for locally targeted cancer awareness intervention programs and serve as a baseline measure against which to evaluate interventions in SSA”

114-115 - “We conducted a community-based cross-sectional survey of women aged 18 years and older in one urban and one rural site in Uganda and SA (total of 4 sites).” How were the sites selected and how much were they representative of the respective countries in terms of socio-economic and demographic variables. Also a brief description of the two countries in terms of number of states, population, etc. may help. The paper seems to generalize the findings for the two countries respectively from the study, which may not be possible if there are widespread differences across different states within each country.

The sites were not selected to be representative of each country. We have added some information on each country and site in the methods section and have amended the conclusion to indicate that the results could inform intervention development in similar settings. 

Methods “Uganda is a low-income country with a population of 34.9 million. Both the urban and rural study sites were in Northern Uganda, the poorest of the four regions in Uganda. SA is a middle-income country with a population of 58.8 million. Our urban SA study site was in the Western Cape Province, one of the wealthier of the nine provinces in the country, whereas our rural site was in the Eastern Cape Province which has very low levels of wealth.”

174 -175 - “We used the same open and closed questions to measure recall and recognition of cervical cancer risk factors (11 items) and symptoms (11 items)” - the meaning conveyed from this line is not clear. One of the limitation in the discussion states that the questions for recall and recognition were asked differently.

We have clarified that we were referring to the AWACAN tool format and have revised this to read “We used the AWACAN tool open and closed question format to measure recall and recognition of cervical cancer risk factors and symptoms . There were 11 closed questions assessing cervical cancer risk factor recognition and 11 questions assessing symptom recognition [15].”

As noted the reason for and limitation of asking the recall and recognition symptom questions slightly differently is described in the discussion section.

206-207 - “Medians and interquartile ranges (IQR) are reported for continuous

variables.” - to be written in past tense

Revised to read “Medians and interquartile ranges (IQR) were calculated for continuous

variables.”

Tables 2,3,4 and 5 – Please mention the test in the footnotes. Based on my interpretation from the description, the places where z value is reported is meant for the Man Whitney U test. Those who have not heard of breast cancer should not be included in symptom identification as the awareness about the symptom if the person is not aware about the disease becomes irrelevant.

We have added the following footnote to each of these tables.

“Where 2 categories of independent variables are compared: z statistic is reported for Mann-Whitney two sample test

Where 3 categories of independent variables are compared: chi-squared statistic is report for Kruskal-Wallis rank test”

We have now clarified in the methods section that we did ask symptom awareness questions even if a participant had not heard of cancer (173-177) .Our symptom recognition questions were framed around whether a symptom is viewed as a sign of something serious (as interpretation of symptoms as potentially serious is likely to drive help-seeking) – having not previously heard of cancer does not preclude this. 

In methodology there is mention of only prompted questions on lay beliefs whereas the results discuss both prompted and unprompted ones. For measuring lay beliefs, unprompted questions may be considered more appropriate. As for the prompted questions, the respondent may not be holding the belief but may choose one from the responses which may be biased (eg - social desirability bias)

The unprompted lay beliefs were obtained from the open questions (e.g. “Please can you name as many things as you can think of that could increase any woman’s chances of getting breast cancer”) We have clarified by adding the following sentence to the methods sections “Additional lay beliefs were identified during unprompted questioning”.

337 Table 4 has a hastag – the footnote is missing

Thank you for pointing this error out, we have removed the hashtag. 

391 “Adapting existing interventions to include additional messaging will be an efficient and cost-effective approach to addressing low breast cancer risk factor awareness.” - This statement should be supported with appropriate evidence.

We have revised this sentence to indicate that this “could be an efficient and cost-effective approach to addressing low breast cancer risk factor awareness.”

406-407 - “Of concern is the finding that many women in rural SA and urban and rural Uganda, did not identify lack of screening as a risk factor for cervical cancer ??” Considering a lack of screening as a risk factor for cervical cancer may not be appropriate as most other risk factors are related to the occurrence of cervical cancer while lack of screening may be a risk factor for delayed presentation of cervical cancer but technically it may be in-appropriate to consider it as a risk factor for occurrence of cervical cancer.

The item related to screening is included in the Cancer Research UK Cervical Cancer Awareness Measurement tool as well as the AWACAN breast and cervical cancer tool. Both tools have been previously validated. Although lack of screening is different to the other aetiological causes, it is associated with a high incidence of cervical cancer.

408 -409 ”Poor screening awareness has also been reported elsewhere in Africa [28,29]. Cervical cancer screening coverage is suboptimal in SSA [30]. In SA, after more than 15 years of a public sector screening program national screening coverage is at 65%, with some provinces having coverage rates as low as 46% [26,31]. In Uganda, there is no national population based cervical cancer screening program. Limited opportunistic screening takes place at national referral and regional referral hospitals when women attend for other reasons. Studies have reported low self-reported cervical screening uptake among community women in different parts of Uganda, ranging from 4.8% to 7.0% [32,33].” - The description of status of screening services in the two countries may be reduced significantly or removed completely, it is not enriching the discussion in the context of the study's findings. Similarly the discussion on HIV and HPV vaccine needs to be reduced significantly.

We have reduced the discussion on screening but have not removed it completely for reasons outlined in our earlier response. The detailed discussion on HIV and HPV were added to the original manuscript following comments from reviewer 2. We have now substantially reduced the discussion on HIV and HPV.

512 – The way in which the questions related to recognition of breast and cervical cancer symptoms were asked appear to be a major limitation of the study. While the methodology described in the study talks about the measurement of awareness of breast and cervical cancer symptoms as one of the major objectives, the way in which the question has been framed “Can you tell me if you think the following could be signs of something serious or that something is wrong, such as breast cancer?” the interpretation and responses would not have been as per the objective. This significantly reduces the confidence of the reader on the findings on the awareness of the symptoms. An interaction with the data collection team to understand how the question was interpreted by most of the respondents may help in addressing this issue.

This limitation is raised in the discussion (516-524). Our recall questions asked specifically about cervical or breast cancer symptoms. However we intentionally worded the recognition questions to acknowledge that interpreting a symptom as something “serious” is likely to drive help-seeking rather than just attribution specifically to cancer. This also enabled us to ask about symptom awareness even among those that had not heard of the specific cancer. 

534 – The conclusion should not be generalised for the entire countries unless the authors are extremely confident of the studied sample being representative of the entire countries. Also a few examples of how exactly can the findings can inform targeted interventions can be of great help to make the description more specific.

We have revised the conclusion to indicate that findings can inform intervention development in similar settings. 

“Our study identified important and specific gaps in community breast and cervical cancer risk and symptom awareness as well as cancer-related lay beliefs in rural and urban settings in SA and Uganda. These results could inform targeted intervention programs in similar urban and rural locations in both countries e.g. media campaigns to increase awareness of the non-lump symptoms for breast cancer. Our findings also provide an important baseline measure for the evaluation of future interventions.”

---

## [Decision Letter · Decision Letter 2]

5 Oct 2020

Mapping awareness of breast and cervical cancer risk factors, symptoms and lay beliefs in Uganda and South Africa

PONE-D-20-06115R2

Dear Dr. Moodley,

We’re pleased to inform you that your manuscript has been judged scientifically suitable for publication and will be formally accepted for publication once it meets all outstanding technical requirements.

Kind regards,

Joel Msafiri Francis, MD, MS, PhD

Academic Editor

PLOS ONE

Additional Editor Comments (optional):

Reviewers' comments:

Reviewer's Responses to Questions

**Comments to the Author**

1. If the authors have adequately addressed your comments raised in a previous round of review and you feel that this manuscript is now acceptable for publication, you may indicate that here to bypass the “Comments to the Author” section, enter your conflict of interest statement in the “Confidential to Editor” section, and submit your "Accept" recommendation.

Reviewer #3: All comments have been addressed

2. Is the manuscript technically sound, and do the data support the conclusions?

Reviewer #3: Yes

3. Has the statistical analysis been performed appropriately and rigorously? 

Reviewer #3: Yes

4. Have the authors made all data underlying the findings in their manuscript fully available?

Reviewer #3: Yes

5. Is the manuscript presented in an intelligible fashion and written in standard English?

Reviewer #3: Yes

6. Review Comments to the Author

Reviewer #3: Dear Authors

Thank you working on the comments and making it appropriate for publication.

We appreciate your efforts.

With good wishes

Abhishek

7. PLOS authors have the option to publish the peer review history of their article (what does this mean?). If published, this will include your full peer review and any attached files.

Reviewer #3: **Yes: **Abhishek Shankar

---

## [Editor Report · Acceptance letter]

13 Oct 2020

PONE-D-20-06115R2 

Mapping awareness of breast and cervical cancer risk factors, symptoms and lay beliefs in Uganda and South Africa 

Dear Dr. Moodley:

I'm pleased to inform you that your manuscript has been deemed suitable for publication in PLOS ONE. Congratulations! Your manuscript is now with our production department. 

Kind regards, 

on behalf of

Dr. Joel Msafiri Francis 

Academic Editor

PLOS ONE